# Effect of Abscisic Acid (ABA) Combined with Two Different Beekeeping Nutritional Strategies to Confront Overwintering: Studies on Honey Bees’ Population Dynamics and Nosemosis

**DOI:** 10.3390/insects10100329

**Published:** 2019-10-01

**Authors:** Nicolás Szawarski, Agustín Saez, Enzo Domínguez, Rachel Dickson, Ángela De Matteis, Carlos Eciolaza, Marcelino Justel, Alfredo Aliano, Pedro Solar, Ignacio Bergara, Claudia Pons, Aldo Bolognesi, Gabriel Carna, Walter Garcia, Omar Garcia, Martin Eguaras, Lorenzo Lamattina, Matías Maggi, Pedro Negri

**Affiliations:** 1Centro de Investigación en Abejas Sociales (CIAS) (IIPROSAM-CONICET), Universidad Nacional de Mar del Plata (UNMdP), Funes 3350, Mar del Plata CP 7600, Argentina; 2INIBIOMA, Universidad Nacional del Comahue, (CONICET), Quintral 1250, Bariloche 8400, Argentina; 3Rocky Mountain Biological Laboratory, PO Box 519, Crested Butte, CO 81224, USA; 4Beekeeper from Azahares del Sudeste association, Instituto Nacional de Tecnología Agropecuaria (INTA), Mar del Plata CP 7600, Argentina; 5Instituto de Investigaciones Biológicas (IIB-CONICET), Universidad Nacional de Mar del Plata (UNMdP), Funes 3350, Mar del Plata CP 7600, Argentina

**Keywords:** *Apis mellifera*, abscisic acid (ABA), nosemosis, overwintering

## Abstract

In temperate climates, beekeeping operations suffer colony losses and colony depopulation of *Apis mellifera* during overwintering, which are associated with biotic and abiotic stressors that impact bees’ health. In this work, we evaluate the impacts of abscisic acid (ABA) dietary supplementation on honey bee colonies kept in Langstroth hives. The effects of ABA were evaluated in combination with two different beekeeping nutritional strategies to confront overwintering: “honey management” and “syrup management”. Specifically, we evaluated strength parameters of honey bee colonies (adult bee and brood population) and the population dynamics of *Nosema* (prevalence and intensity) associated with both nutritional systems and ABA supplementation during the whole study (late autumn-winter-early spring). The entire experiment was designed and performed with a local group of beekeepers, “Azahares del sudeste”, who showed interest in answering problems associated with the management of honey bee colonies during the winter. The results indicated that the ABA supplementation had positive effects on the population dynamics of the *A. mellifera* colonies during overwintering and on the nosemosis at colony level (prevalence) in both nutritional strategies evaluated.

## 1. Introduction

Pollinators play a key role in the functioning of ecosystems and the conservation of biodiversity [1]. Bees comprise one of the main families of pollinators with wide distribution all over the world [2], including native bees and honey bees as the most important pollinators of crops and natural landscapes [3,4].

Colony losses and colony depopulation of the European honey bee *Apis mellifera* have been reported for several years as a phenomenon in some regions of the world [5,6,7,8]. Particularly, in USA colony losses have reached levels up to 50% [9]. These drastic declines could lead to the loss of pollination services generating significant negative effects that could affect the conservation of wild plant communities, crop production, and therefore, food safety [10].

Several factors have been identified as the main drivers of the loss and depopulation of bees. Reports show that the interactions between pathogens and environmental and anthropogenic stressors are acting synergistically to negatively impact bee populations [10,11,12,13,14].

In temperate climates, beekeepers have reported high colony losses and colony depopulation during overwintering [7,15]. The winter is one challenge for bees’ colonies in temperate regions, such as Argentina, which is characterized by having a very cold season. So far, these cold temperatures have been investigated as a factor that can trigger overwintering changes in bees [16,17]. Furthermore, previous studies have demonstrated that levels of *Varroa*, viruses, *Nosema*, and geographic location, among others, are correlated with winter colony losses and colony depopulation [7,18,19]. It is for these reasons that in temperate climates good management practices of honey bee colonies is essential in order to increase overwintering survival.

One of the most common pathogens of *A. mellifera* is *Nosema*, a microsporidian intracellular parasite that infects the midgut epithelium of honey bees and causes a disease called “nosemosis” [20]. Two species of *Nosema* are able to induce the disease in honeybees, *Nosema apis* and *Nosema ceranae* [21,22], but it has been suggested that the effects of *N. ceranae* on bee colony health are more severe compared to infections by *N. apis* [23]. Higes et al. [18] showed that a natural *N. ceranae* infection caused declines in honey bee populations in Spain, establishing a direct correlation between nosemosis and the death of *A. mellifera* colonies under field conditions. In Argentina (Buenos Aires province), a genetic study reported that all bee samples analysed from 38 apiaries were positive for the presence of *N. ceranae* while only two apiaries were positive for the presence of *N. apis* in co-infection with *N. ceranae* [24]. These results show that *N. ceranae* is widely distributed throughout Argentina.

In Argentinean temperate climates, the application of an antibiotic, fumagillin, is recommended as an effective treatment for the control of the nosemosis during autumn, since it would maintain low levels of spores to avoid the peak of the development of the disease that occurs from the end of winter to the early spring [25]. Although fumagillin has been proposed to control nosemosis in bee colonies [26], its use is illegal to use in honey bee colonies in many countries, mainly in Europe, due to the residuals in honey [27], which are toxic to humans [28]. In past years, the search for natural alternatives to control nosemosis through integrated pest management (IPM) has arisen. One example is the application of bacterial metabolites on bee colonies in field conditions, which have been shown to negatively impact *N. ceranae* [29,30,31].

In recent years new reports have shown evidence that plants’ molecules, found in nectar and/or pollen, play an important role in pollinators health through immune system strengthening [32,33,34]. Naturally, pollinators may benefit from the consumption of nectars and pollens, rich in plant-originated compounds, through a phenomenon known as self-medication [33]. For example, Mao et al. [35] identified that p-coumaric acid, a main component of the pollen grain wall, stimulates detoxification mechanisms by expression gen family P-450 in *A. mellifera*, increasing the bees’ tolerance to pesticides. In 2015, Richardson et al. [36] tested the effects of feeding bumblebees with eight different chemicals found in nectar, and they observed that four of these natural molecules tested (anabasine, catalpol, nicotine and thymol) strongly reduced the levels of *Cri*th*idia bombi*, an intestinal parasite. In a similar work, but with broader approaches, Baracchiet et al. [37] showed that nicotine, a secondary metabolite found in nectar most commonly in *Solanaceae* and *Tilia* species, is used by parasitized bumblebees as a source of self-medication. They demonstrated that bumblebees parasitized by *Cri*th*idia bombi* modify their diet preference and foraging behaviours, delaying the development of the infection. Recently, Palmer-Young and co-workers [38] reported that the consumption of phytochemicals from nectar and pollen significantly increased antimicrobial peptide expression in old honey bees after 7 days of consumption and reduced levels of Deformed wing virus (DWV) in young honey bees over the short-term (<24 h).

During the past years, our group has focused on the role that abscisic acid (ABA), a natural component that is present in nectar, honey, and pollen [39,40] as well in honey bees [39,41] plays an important role in bees’ health. ABA is a sesquiterpenoid hormone that play important roles, mainly in higher plants, in many cellular processes including seed development, dormancy, germination, vegetative growth, and environmental stress responses [42]. Negri et al. [41] evaluated the effects of ABA on the bees’ immune responses in field conditions by analysing the effects of this molecule over the performance of small *A. mellifera* colonies throughout the winter season. They observed that ABA had an important effect at the individual level, stimulating the cellular and humoral innate responses, and at a colony level where populations of adult bees supplemented with ABA were approximately 70% larger than control populations after winter. In another study, Ramirez et al. [17] showed that supplementing bees’ diets with ABA prevented low survival rate and accelerated adult emergence of in vitro-reared honeybee larvae exposed to low temperatures (25 °C). In this work, the authors also showed that ABA supplementation enhanced the expression of genes involved in metabolic and stress responses.

However, the question regarding the effects of ABA on Langstroth colony sizes still remained unanswered. In this study we used an experimental approach designed to evaluate the impacts of ABA supplementation on brood and adult honey bee populations from colonies kept in Langstroth hives, under two beekeeping strategies that differ in the way they manage bee nutrition to confront overwintering: one where bees are fed honey during the winter and the other where bees are fed with enough syrup (sugar:water) so that all cells in the brood area of the hive are full. This second strategy, called “blocking”, is utilised by beekeepers before the beginning of the winter to prevent queens from egg laying during cold climatic conditions. In addition, we decided to study the population dynamics of *Nosema* (prevalence and intensity) in bee colonies associated with both nutritional systems throughout the entirety of the study (late autumn-winter-early spring). The entire experiment was designed and performed with a group of beekeepers, “Azahares del sudeste”, who also showed interest in answering our two questions. This study was successfully accomplished in full interaction and collaboration with these beekeepers.

## 2. Materials and Methods

### 2.1. Location Study

Field trials were carried out in an apiary commercial located in Miramar, Buenos Aires Province, Argentina (38°14’41.42” S, 57°49’2.09” W). The apiary belonged to the group of beekeepers “Azahares del Sudeste”.

### 2.2. Honey Bees

Field research was carried out with a local *A. mellifera* ecotype (A*. mellifera mellifera* × *A. mellifera ligustica*) [43], with queens sharing the same genetics, obtained from mother colonies of the queen bee farm belonging to the group of beekeepers “Azahares del Sudeste”. A total of 20 bee colonies were used (5 per treatment), kept in standard Langstroth-size hives. To confront overwintering (see in Appendix A the temperature fluctuations from autumn to the end of winter), two different beekeeping practices that differed on the manner of feeding were applied: “honey management”, where bees are fed with honey (this is the typical management strategy used by “Azahares del sudeste” as an overwintering strategy) and “syrup management”, where bees are fed with sugar syrup (2:1, sugar:water, *v*:*v*) until the cells in the brood area are full. This is usually done by beekeepers to feed the bees and simultaneously decrease and/or avoid the egg laying of the queen during winter. Before the beginning of the trials, bee colonies were prepared according to the management characteristics of each productive system. In addition, previously to the start of the experiment, in order to honey bees’ colonies standardize, the number of combs designated for adult bees, brood, and food storage (both honey and pollen) was quantified in every one. In this way, 4 groups of 5 hives each as were formed as similar as possible in respect to these variables.

### 2.3. Phytochemical

S-Abscisic acid (ABA) was purchased from FandaChem (S-ABA, S-(+)-Abscisic Acid, CAS N° 21293-29-8, www.fandachem.com, Hangzhou, Zhejiang, China). The final concentration of ABA dissolved in syrup (2:1, sugar:water, v:v) was 50 µM. Previous to the field experiment, assays were performed to test the toxicity of the compound under controlled laboratory conditions (CIAS, UNMdP). The results indicated that the compound is not toxic to adult bees (data shown in Appendix A).

### 2.4. Treatments

Experiments were performed between April (autumn) and September (spring) 2017. The treatments were conducted as follows:Control syrup (CS): Five (5) *A. mellifera* colonies integrated in one (1) Langstroth unit (breeding chamber) were fed with 2 L of sugar syrup 2:1(sugar:water, *v*:*v*) weekly, using a Doolittle feeder, until the breeding chamber was full (blocked) of syrup (4 total applications between May 8th to May 29th).ABA syrup (AS): Five (5) *A. mellifera* colonies integrated in one (1) Langstroth unit (breeding chamber) were fed with a supplementary diet of 2 L sugar syrup 2:1 (sugar:water, *v*:*v*) + ABA 50µM (dissolved in the syrup) weekly, using a Doolittle feeder, until the breeding chamber was full (blocked) of syrup (4 total applications between May 8th to May 29th).Control honey (CH): Five (5) *A. mellifera* colonies integrated into two (2) Langstroth units, were organized in the upper chamber with 3 combs of honey + 7 empty combs. The honey of the combs corresponded to the one produced by the colonies in the latest season before this experiment. In one of the empty combs that was placed in-between the combs with honey, we applied 0.2 L of sugar syrup 2:1(sugar:water, *v*:*v*), two (2) times per week (Monday and Friday), for 4 weeks (8 total applications comprised between May 8th to June 2nd).ABA honey (AH): Five (5) *A. mellifera* colonies integrated into two (2) Langstroth units, were organized in the upper chamber with 3 combs of honey + 7 empty combs. The honey of the combs corresponded to the one produced by the colonies in the latest season before this experiment. In one of the empty combs that was placed in-between the combs with honey, we applied 0.2 L of sugar syrup 2:1(sugar:water, *v*:*v*) + ABA 50µM (dissolved in the syrup) two (2) times per week, for 4 weeks (8 total applications comprised between May 8th to June 2nd).

### 2.5. Colony Population Dynamics

The progress of the honey bee colonies (controls and supplemented with syrup plus ABA) was monitored at six time-points throughout the entire experiment: May 9th, May 23th, June 6th, June 13th, August 2nd and September 18th (from autumn to the end of winter). The colony strength parameters were measured according to the subjective mode [44]. The parameters to quantify the general state of the colonies during the evaluations were as follows: number of combs covered with adult bees (grade), and number of combs covered with brood.

### 2.6. Nosema Quantification

Samples of forager bees from each colony were taken prior to the first treatment application: (a) time-point T0 (May 8th); (b) time-point T2 (June 2nd); (c) time-point T3 (August 2nd) and (d) time-point T4 (September 18th). Samples were taken using forager bees because they are more frequently infected compared to younger bees and they were collected midday at the same timepoint to be comparable across experiments [45]. The entrance of every beehive was closed with foam so that foraging bees piled up at the entrance and a representative sample group of approximately 100 individuals could be collected and put into a jar of 70% ethanol [20]. According to the methodology described by Fries et al. [20], to determine Prevalence (number of bees infected with *Nosema*/n ° bees totally examined) and Intensity (number of *Nosema* spores/parasitized bee examined) the abdomens of 21 bees from each sample group were individually homogenized in 1 mL of distilled water and checked for the presence of *Nosema* spores using a Neubauer count camera (BOECO^®^, Hamburg, Germany) under an optic microscope (Leica^®^ DM 500, Wetzlar, Germany) (480×).

### 2.7. Statistics

#### 2.7.1. Colony Population Dynamics

To evaluate the effects of treatments and time on honey bees’ colonies development was performed Two-Way ANOVA (*p* < 0.05) followed by a post-hoc analysis of Holm-Sidak (*p* < 0.05) for multiple comparisons, using the SigmaPlot software (version 11.0, SYSTAT SOFTWARE^®^, San Jose, CA, USA). The tests of Shapiro-Wilks (*p* < 0.05) and Levene’s (*p* < 0.05) were used to verify normality and homoscedasticity assumptions. All the data set were normally distributed.

#### 2.7.2. *Nosema* Prevalence and Intensity

To evaluate the effects of the different dietary treatments (i.e., control syrup, ABA syrup, control honey and ABA honey) on *Nosema* intensity and prevalence through the overwintering period we used generalized linear mixed-effects models, given the nature of the response variables and the structure of the hierarchical sampling design (i.e., repeated measurements of bees per colony) [46,47]. Since the response variables for *Nosema* intensity analysis follow a discrete nature (i.e., number of spores per bee), the model assumed a Poisson error distribution with a *log* link function. Because the response variable for *Nosema* prevalence analysis follows a Bernoulli trial process (i.e., bees infected vs non-infected), the model assumed a Binomial error distribution with a *logit* link function. Dietary treatment (i.e., categorical variable with four levels) and its interaction with time of the year (i.e., before vs after overwinter) were included as fixed effects, and “colony” as random effects. Data analysis was carried out using the *lmer* function from the *lme4*package [48], and we made multiple comparison of means with a Tukey post-hoc test using the *lsmean* function from the *lsmeans* package [49] of R software (version 2.15.1).

## 3. Results

### 3.1. Effects of ABA Dietary Supplementation on Colony Population Dynamics

The results obtained regarding the effects of ABA supplementation on the development of the A. mellifera colonies are shown in Figure 1 and Figure 2. The first sampling point (May 9th) showed in Figure 1a, revealed that the control group associated with the honey management started the experiment with slightly lower levels of adult bees than the ABA-supplemented group (6.6 vs 8 adult bees combs respectively) However, this initial difference was not significant (two-way ANOVA, *p* = 0.002; Holm-Sidak post-hoc, *p* = 0.137). Then, in sampling point June 6th, the adult bee levels of the control and ABA groups were similar (7.4 vs 7.2 adults bees combs respectively) (two-way ANOVA, *p* = 0.002; Holm-Sidak post-hoc, *p* = 0.830). From this point on, the levels of adult bees in the ABA-supplemented group began to increase, separating from the control group. And finally, as can be seen in Figure 1a, ABA supplementation associated with honey the feeding strategy resulted in a statistically significant difference of 3.3 more combs covered in adult bees at the end of winter compared to the control (two-way ANOVA, *p* = 0.002; Holm-Sidak post-hoc, *p* < 0.001). Figure 1b shows that in the first sampling point (May 9th) of the syrup feeding assay the levels of adult bee in the ABA-supplemented group and the control group were similar (7.8 vs 7.3 adult bees combs respectively) (two-way ANOVA, *p* = 0.001; Holm-Sidak post-hoc, *p* = 0.512). Then, from sampling point August 2nd on, the levels of adult bee in the ABA-supplemented group began to increase, separating from the control group. And finally, as can be seen in Figure 1b, ABA dietary supplementation associated with the syrup feeding strategy resulted in a statistically significant difference of 3.67 more combs covered in adult bees at the end of winter compared to the control (two-way ANOVA, *p* = 0.001; Holm-Sidak post-hoc, *p* < 0.001). The brood population of the colonies also varied through the trial in both field operations. The first sampling point (May 9th) showed in Figure 2a, revealed that the ABA-supplemented group associated with the honey management started the experiment with lower levels of brood than the control group (2 vs 3.3 brood combs respectively). This resulted in significantly lower levels of brood before winter in the ABA group compared to the control at sampling points May 23th (two-way ANOVA, *p* = 0.045; Holm-Sidak post-hoc, *p* = 0.026) and June 6th (two-way ANOVA, *p* = 0.045; Holm-Sidak post-hoc, *p* = 0.050). Interestingly, ABA supplementation reversed this initial decline showing a significative increase of brood population (two-way ANOVA, *p* < 0.001; Holm-Sidak post-hoc, *p* < 0.001) through winter while the control group did not (two-way ANOVA, *p* < 0.001; Holm-Sidak post-hoc, *p* = 0.626). Though not significative, this resulted in higher levels of brood in the ABA group in comparison to the control at the last sampling point (September 18th) (two-way ANOVA, *p* = 0.045; Holm-Sidak post-hoc, *p* = 0.068). As seen in Figure 2a, the control group started with 3.3 and ended with 4.6 brood combs (meaning an increase of 1.3 combs through the winter) while the ABA group started with 2 and ended with 5.9 brood combs (revealing an increase of 3.9 combs during winter). Figure 2b shows that ABA dietary supplementation associated with the syrup feeding strategy resulted in a statistically significant difference of 2.57 more combs covered in brood at the end of winter compared to the control (two-way ANOVA, *p* < 0.001; Holm-Sidak post-hoc, *p* = 0.002). These results suggest that the dietary supplementation of ABA to Langstroth-size bee colonies has positive effects on the abilities of both adult and brood populations to effectively overwinter.

### 3.2. Effects of ABA Dietary Supplementation on Nosema Levels

#### 3.2.1. Nosemosis at Colony Level

While Control groups increased the percentage of parasitized bees by Nosema for the two types of management applied (syrup or honey) at the end of the winter, ABA groups showed no increased (Figure 3). Prevalence of Nosema showed that there were no significant differences between the treatments control syrup(CS), control honey (CH), ABA syrup (AS) and ABA honey (AH) at the start of the trial (time-point T0, May 8th) (Tukey post-hoc test, *p* > 0.05 in all cases), indicating that all bee colonies started with a similar parasitic load. By comparing the prevalence of Nosema within each treatment over time from T0 (“autumn”, May 8th) to T4 (“end winter”, September 18th), we obtained the following results: In T4 the colonies of bees belonging to the control groups CS and CH were statistically different from T0, revealing an increase in the prevalence of Nosema at end of winter (Tukey post-hoc test, *p* < 0.001 in both cases). On the contrary, in bee colonies belonging to the AS and AH treatments, the prevalence of Nosema at the end of winter were not significantly different from the prevalence in T0 (Tukey post-hoc test, *p* = 0.78 and *p* = 0.28, respectively).

#### 3.2.2. Nosemosis at Individual Level

Intensity of Nosema (n° spores per infected bee) through the winter was similar for the four treatments applied in the field trial (Figure 4). For each feeding treatment (ABA syrup and ABA honey, control syrup, control honey,), the spore loads per bee increased over time, between time-point T0 (May 8th) and time-point T4 (September 18th), showing statistically significant differences (Tukey post-hoc test, *p* < 0.001 in all cases) (Figure 4).

## 4. Discussion

Nutritional deficiencies can affect the immune responses of honey bees and accelerate the spread of disease among nest mates, increasing pathogen levels and reducing adult longevity and survival [12,50]. When bees collect pollen and nectar from plants they do not only forage for proteins and carbohydrates, but also forage for associated plant-derived products (e.g., secondary metabolites, phytohormones). These molecules impact individual bees’ physiologies and in turn can shape the health of entire colonies [33]. These plant-derived components are important for self-medication at both the individual and colony levels and can both decrease diseases and enhance immunity of bees [33,38].

Our study evaluated the effect of the supplementation of abscisic acid (ABA), a plant-derived molecule, on the ability of Langstroth-size colonies of *Apis mellifera* to overwinter. The effect of ABA was evaluated under two beekeeping strategies that differ in the way they manage bee nutrition to confront winter: “honey management” and “syrup management”, which feed the bees with honey or sugar-water syrup respectively. Our results indicated that ABA had positive effects on the population dynamics of overwintering *A. mellifera* colonies and on the prevalence of nosemosis at colony level (one of the main diseases affecting *A. mellifera*) in the two types of productive management evaluated.

We applied a standard ABA concentration of 50 µM because: (a) it is within the range of previously reported ABA concentrations found in nectar, pollen [40], honey and bees [17,41] and (b) this concentration has been shown to have positive effects on the immunity of bee larvae reared in the laboratory in response to various stress conditions [17,51]. Further, the concentration of ABA found in local honey are very low (around 0.0008·μmol/g, data obtained by our group) in comparison with the concentration found in other honeys, like for example the monofloral honey of strawberry tree (*Arbutus unedo* L., around 1.5 μmol/g) [40]. The concentration of ABA used in this work to supplement bee colonies under both managements was considerably higher than the levels detected in local honey (expressed as 50 μM, which means around 0.04 μmol/g) but not as high as the concentration found in *A. unedo* L. honey. This reinforces the importance of supplementing bee’s nutrition with ABA in both managements (honey or syrup) evaluated in this work. Before this study, in the field, ABA had only been studied in honey bees reared in “minihives” (small experimental units) [41]. The results obtained through this work are consistent with those in Negri et al. [41] regarding the effect of ABA on bee population through overwintering. In our study, the size of the colonies of *A. mellifera* overwintering was associated with the current productive system (beekeeping using Langstroth-size colonies). In agreement with the results of Negri et al. [41], we show that the ABA supplemented colonies retained their initial adult population after the winter for the two types of nutritional strategies evaluated.

In this study, we also observed a positive effect of ABA supplementation on brood population throughout the overwintering period. These results are supported by the results reported by Ramirez et al. [17] where the authors showed that supplementing bees’ diets with ABA prevented low survival rates and accelerated adult emergence of in vitro-reared honey bee larvae exposed to low temperatures.

We must mention that within the group associated with the honey management, no significant differences were obtained at the end of winter (September 18th) between ABA honey (AH) and control honey (CH) treatments. However, if we analyze the evolution of brood growth within both groups throughout the experiment the positive effects of ABA should be noted. At the beginning of the experiment (May 9th), the number of combs covered with bee brood in the AH was lower than the CH (see Figure 2a), although without significant differences. For the next two time-points (May 23th and June 6th) the differences in the combs covered with bee brood between both treatments were significant. However, again in the fourth time-point (June 13th), the differences between AH and CH began to decrease, becoming statistically not significant. From this time-point until the end of the trial, the levels of bee brood in the treatment with ABA (AH) remained above the control (CH), reversing the differences found at the beginning of the experiment. This can be seen in Figure 2a, where the control group started with 3.3 and ended with 4.6 brood combs (meaning an increase of 1.3 combs through the winter) while the ABA group started with 2 and ended with 5.9 brood combs (revealing an increase of 3.9 combs during winter). These results highlight two important findings: (a) the positive effects of ABA treatment over honey bee colonies’ brood growth, and (b) the relevance of standardizing the population of the colonies before the start of these types of studies. In our case, we tried to standardize all bee colonies as homogeneously as possible, but some differences still existed between the colonies. Although these differences were not statistically significant at the beginning of the experiment, they can still introduce variations that should be taken into consideration and may not necessarily be related to the aims of the study. A honey bee colony is a eusocial superorganism and its status at the start of an experiment affects the way in which it responds to different treatments or stress factors.

Considering that honey has important properties for the self-medication of bees due to the contribution of various compounds, such as antimicrobial compounds (e.g., antimicrobial peptide bee defensin-1) [52], major royal jelly protein 1 [53], secondary plant metabolites including alkaloids and phenolic acids (e.g., caffeic acid, p-coumaric acid) [32,35], and phytohormones [54], we thought it was interesting to compare both types of conditions (with or without honey) adding the supplementary diet with ABA to honey bee colonies. Due to these characteristics of honey that influence the bee’s health through pharmacophagy (resulting from the direct consumption) [33], we expected to not find differences between bee colonies that only received honey with those who received honey + ABA. However, both treatments that received ABA (honey + ABA and sugar syrup + ABA) resulted in increased population sizes compared to control treatments (only honey or syrup sugar) at the end of winter.

Within the context of bees’ nutrition and overwintering, Ricigliano and co-workers [55] studied the effects (pre and post-winter) of forage environment in apiaries close to agricultural or non-agricultural landscapes (Conservation Research Program (CRP) lands) on the colonies’ strengths. In their work, the authors highlighted the importance of pre- and post-winter time points as critical periods for assessing the health of the colonies. The latter is in great agreement with the experimental design of this work and, at the same time, the results support that ABA can be applied in the pre-winter and considered as a nutritional supplement for bee colonies.

Regarding ABA and its impact on *Nosema* levels, two types of monitoring of the parasite were selected: prevalence (colony level) and intensity (individual level). There are reports that *Nosema* infections do not cause dead honey bee colonies, except in Spain [18,56], but infections almost certainly weaken them and result in decreases of productivity [57]. For example, Botías et al. [58] reported that *N. ceranae* infection was highly pathogenic for honey bee colonies, causing significant reductions in colony size, brood rearing, and honey production. Other studies have concluded that the nosemosis can synergize with other risk factors including environment, pesticides, viruses, diet and *Varroa* [59,60,61].

Our study showed that the supplementation with ABA in both types of winter management strategies evaluated had effects on the dynamics of *Nosema* populations at the colony level. In concordance with previous reports [18,30,58], our results indicated that the levels of parasitized bees increased significantly in honey bee colonies without ABA supplementation. In contrast, when the bees’ diet (honey or sugar syrup) was supplemented with ABA the levels of parasitized individuals did not increase significantly. These results suggest that ABA could suppress the development of nosemosis at a colony level throughout the overwintering period. We also observed that the variation around the mean values increased in the ABA treatments in comparison with the control. This means that colonies responded to the ABA supplementation in a more variable way, while all the untreated colonies increased the levels of parasitized individuals similarly.

Our results indicated that *Nosema* spore loads of infected bees were increasing throughout the winter in all dietary treatments. As previously reported [18,30,61], our results obtained indicated that the levels of *Nosema* infection development at the individual level (intensity) increased significantly throughout winter in honey bees from colonies without ABA supplementation. When the bees’ diet (honey or sugar syrup) was supplemented with ABA the intensity of *Nosema* also increased significantly throughout the winter. However, as shown in Figure 4, the magnitude of the change in the treatments that received ABA was lower compared to the respective controls.

Previous works indicated that the number of spores per parasitized bee (intensity) is not the best parameter to measure colony health, due to the strong variations in spore counts that can be found between different naturally infected colonies of bees [18,45]. The mean rate of infected honey bees (prevalence) seems to be the most reliable method to evaluate the health status of a colony [18,45,58]. Therefore, the results obtained in the proportion of parasitized bees (colony level) could indicate a stimulation of the healthier state of the bee colonies that had been supplemented with ABA to confront winter. However, due to the strong variation that exists in spore counts [18,45], more studies should be performed in order to strengthen these results. For example, longer term experiments implementing ABA supplementation in honey bee colonies (e.g., seasonal: autumn, winter, spring, summer) could give appreciable results and it could give a view of how parasite dynamics change across a broader time scale.

In 2016, Jack et al. [62] showed that honey bees’ diets with higher pollen quantities increase *N. ceranae* intensity but they found no significant differences in prevalence among different pollen diet treatments. Similar results were obtained by Porrini et al. [63] and Basualdo et al. [64], among others. This phenomenon may be due to the fact that *N. ceranae* is highly dependent on host nutritional status for its development, suggesting that the microparasite uses amino acids from its hosts. [61,65]. In the future, controlled laboratory experiments should be conducted to search for a possible mechanism that could be associated with the effects of ABA over *Nosema* development in honey bees (e.g., bioassays that inoculate bees individually with standardized concentrations).

As it was mentioned before, in temperate climates, low temperatures can negatively influence the survival of honey bee colonies during the winter [7,15]. Taking into account that the dynamics of nosemosis shows its highest degree of development in late winter/early spring [18,25], a good autumnal management could represent the key to mitigate the negative effects of the disease. In this sense, supplementing beehives with ABA during autumn could represent a good nutritional strategy in the two types of productive management evaluated: “honey” and “syrup” managements.

## 5. Conclusions

The results obtained in this work suggest that ABA enhanced Langstroth-size honey bee colonies’ strengths associated with differences in population dynamics (stable levels of adults and brood) and the changes levels of prevalence of *Nosema* obtained overwintering. These results support all previous reports of the effects of ABA in honey bees [17,41,51] adding new evidence to suggest that this phytochemical is a natural alternative to be used in environmentally friendly strategies associated with the integrated management of bee colonies.

## Figures and Tables

**Figure 1 insects-10-00329-f001:**
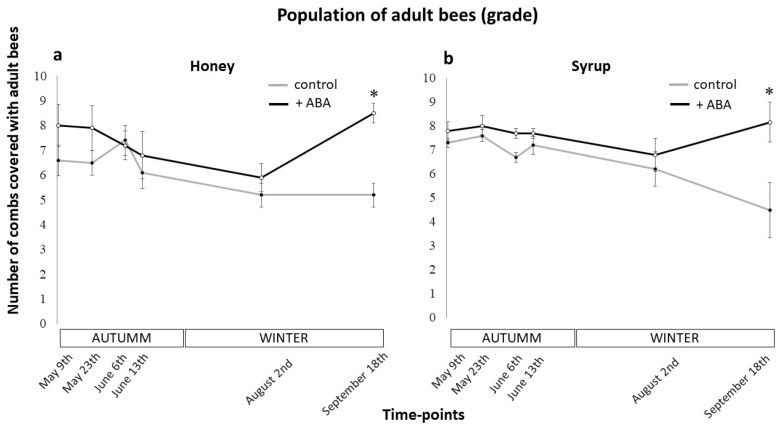
Effect of abscisic acid (ABA) on the dynamic of adult bee population during the field trial. The progress of the adult bee population of the honey bee colonies supplemented with 50 µM ABA (+ABA) or not (control) corresponding to the honey (**a**) or syrup management (**b**) was monitored six times throughout the experiment: May 9th, May 23th, June 6th, June 13th, August 2nd and September 18th (from autumn to the end of winter). A total of 20 bee colonies were used (5 per treatment), kept in standard Langstroth-size hives. Dots indicate mean values, while bars ±1 SE. The asterisks indicate significant differences (two-way ANOVA, *p* < 0.05, Holm-Sidak post-hoc, *p* < 0.001).

**Figure 2 insects-10-00329-f002:**
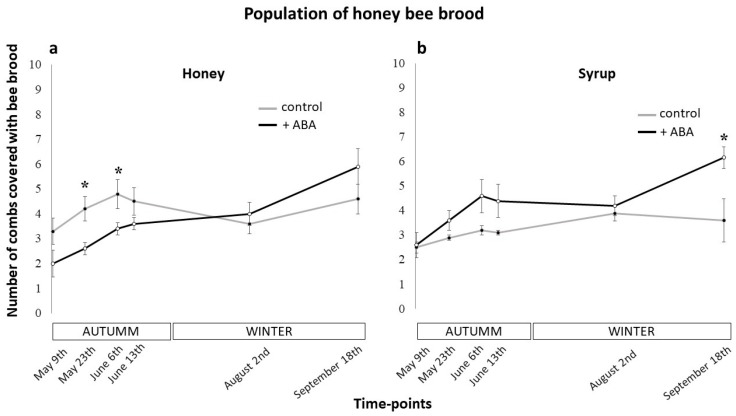
Effect of ABA on the dynamic of brood bee population during the field trial. The progress of the brood bee population of the honey bee colonies supplemented with abscisic acid (ABA) 50 µM (+ABA) or not (control) corresponding to the honey (**a**) or syrup management (**b**) was monitored six times throughout the experiment: May 9th, May 23th, June 6th, June 13th, August 2nd and September 18th (from autumn to the end of winter). A total of 20 bee colonies were used (5 per treatment), kept in standard Langstroth-size hives. Dots indicate mean values, while bars ±1 SE. The asterisks indicate significant differences (two-way ANOVA *p* < 0.05, Holm-Sidak post-hoc, *p* < 0.05).

**Figure 3 insects-10-00329-f003:**
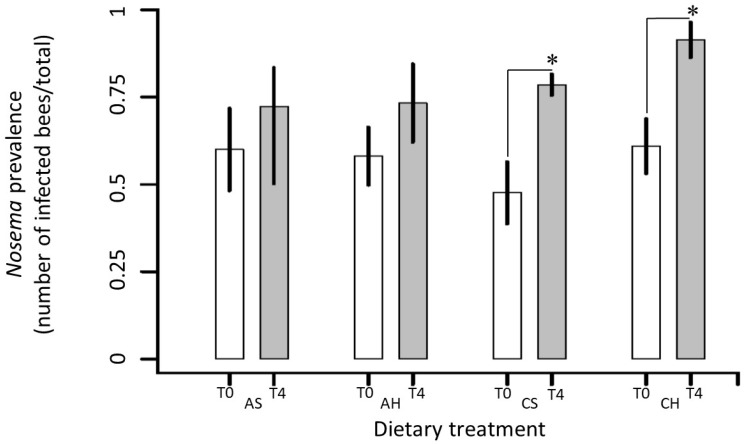
Changes in Nosemosis at colony level. *Nosema prevalence* of forager bees from the two types of hive management fed with ABA syrup (AS), ABA honey (AH), control syrup (CS) and control honey (CH) before (T0: May 8th, white bars) and after (T4: September 18th, grey bars) overwinter. A total of 105 bees per treatment (21 individuals from the 5 colonies comprising each sample group) were analyzed individually. In T4 the colonies of bees belonging to the control groups CS and CH were statistically different from T0, registering an increase in the prevalence of *Nosema* at end of winter. Thick and thin bars show mean ±1 SE of *Nosema* prevalence. The asterisks indicate significant differences (Tukey post-hoc test, *p* < 0.05).

**Figure 4 insects-10-00329-f004:**
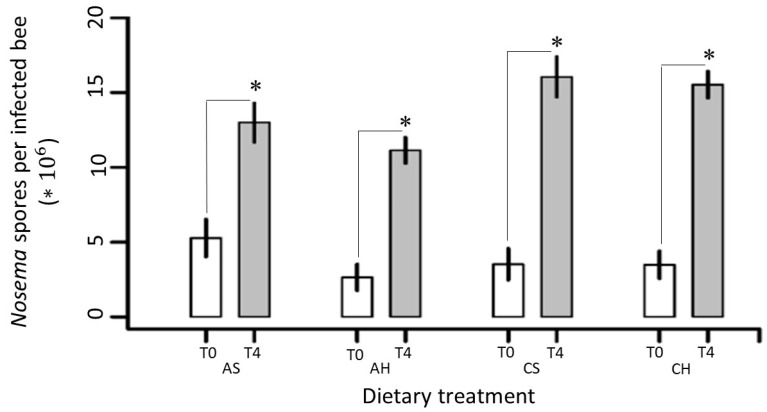
Changes in Nosemosis at individual level. *Nosema* spores per infected bee in hives fed with ABA syrup (AS), ABA honey (AH), control syrup (CS), and control honey (CH) before (T0: May 8th, white bars) and after (T4: September 18th, grey bars) overwinter. A total of 105 bees per treatment (21 individuals from the 5 colonies comprising each sample group) were individually analyzed. In all dietary treatments, the spore loads per bee increased over time, between time-point T0 (May 8th) and time-point T4 (September 18th), resulting in statistically different spore loads within each dietary group (Tukey post-hoc test, *p* < 0.05). Thick and thin bars show mean ±1 SE of *Nosema* spores per bee (in 10^6^ scale).

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
