# Peer review of "Effect of Abscisic Acid (ABA) Combined with Two Different Beekeeping Nutritional Strategies to Confront Overwintering: Studies on Honey Bees’ Population Dynamics and Nosemosis"

_insects, 2019, doi:10.3390/insects10100329_

Round 1

Reviewer 1 Report

Dear Authors,

In the manuscript titled “Effect of abscisic acid (ABA) combined with two different beekeeping nutritional strategies to confront winter: Studies on honey bee’s population dynamics and nosemosis” Szawarski and colleagues studied whether a dietary supplement with abscisic acid (ABA) improved overwintering success in honeybee hives. The question is interesting and relevant as ABA is a natural plant hormone present in nectar, honey and pollen of several plant species pollinated by honeybees and other relevant pollinators. It is already known that ABA has an important role for honeybees’ health and several studies have been already conducted in the past on this topic. In particular, as explained by the authors, it is already well known that ABA has a beneficial role both at the individual level, stimulating the cellular and humoral innate response, and at a colony level. However, previous studies have been conducted on experimental small colonies (mini hives). Therefore, this study aimed at clarifying whether ABA has a positive impact on the health in colonies kept in Langstrong hives, which is a relevant question as this type of beehive is the most used in the world.

The study is well done, quite straightforward and results clear enough and in line with previous findings. Therefore it deserves publication, after few improvements. I only have a major point that authors have to address about the statistical reports provided in the Results section. Moreover, I also have few additional suggestions and advices that might improve the Ms.

Main point

In the Result section paragraph 3.1 (Lines 204, 207, 2010) authors only reported the post hoc values, but they must report the P value of the entire models (I.e. the p value of the main factor osr the interaction when relevant). The same holds for lines 235, 240, 242) paragraph 3.2.1. In all cases the authors should report the exact value not just >/< of 0.05.

Minor points:

-The introduction might be a little bit more concise and the English of this section (and the Discussion) should be improved a bit.

- Lines 50-55 could be reduced at two short sentences. Line 75-83 might be deleted. In my opinion, the short paragraph (lines 84-90) might be extended a little bit to make the introduction more appealing and informative. The subject introduced in this paragraph is indeed relevant to the main topic of the manuscript. Moreover, some important citations are missing:

Palmer‐Young et al 2011 doi: 10.1093/jee/tox193

Baracchi et al 2015 doi: 10.12688/f1000research.6262.3 (also in Line 274),

Palmer‐Young et al 2017 DOI: 10.1002/ece3.2794

In lines 136-137 the authors reported the ABA concentrations in µM and then in %. Please reconcile these units.

In figures 3 and Figure 4, the four groups are named differently from the text (I.e. SA instead of AS, AH instead of HA, CS and CH instead of SC and HC). Please amend it.

Author Response

Dear,

Thanks for everything,

Nicolás Szawarski

Reviewer 2 Report

This is a description of an important study pertaining to honey bee colony health and viability. Nosema ceranae is a worldwide problem for beekeepers, so any contribution to a solution is most welcome. I recommend some minor changes and clarifications.

Lines 110, 125, 145 etc    “Blocking” is a term not understood by everyone. It would be better to say something like “feeding enough syrup or honey so that all cells in the brood area of the hive are full”.

Lines 128, 135 etc. Is “syrup” sucrose dissolved in water? At what concentration? Or is it corn syrup?

Line 136                “… the toxicity of 90% ABA …” Is not clear. I’m sure that the bees were not fed 90% ABA. Does this mean “ .. a preparation of 50 uM ABA (made from 90% ABA) in syrup “ ?

Line 179               Please provide the vendor and the location of the vendor (city and country) for the Neubauer camera.

Line 232                “(Figure 3)” should follow the sentence that ends with “… the end of winter.”   

Line 247                “(Figure 4)” should follow the sentence that ends with “…in the field trial.           

Figure 3                 Are these data % or proportion of infected bees? 0.75% equals 75 bees per 10,000.

Line 252                A better title for the caption would be “Changes in Nosemosis at colony level.” because that is the essence of the graph.

Line 257                This should read “… groups CS and CH …”

Line 261                A better title for the caption would be “Changes in Nosemosis at individual level.”

Author Response

(The authors gave the same response as above.)

Round 2

Reviewer 1 Report

Dear Editor,

The manuscript has been accurately revised and authors addressed all the points raised by me. Statistics is now clear and the results exhaustive. In my opinion it can be accepted for publication in Insects in its present form.

Best Regards,

David 

Author Response

Dear,

We are very happy to know that the reviewer was satisfied by our responses to her/his comments. We really appreciate her/his work, it helped us to improve the quality of the present study.

Thanks for everything

Best regards,

Nicolás Szawarski and co-workers